# Challenges and Opportunities in the Process Development of Chimeric Vaccines

**DOI:** 10.3390/vaccines11121828

**Published:** 2023-12-08

**Authors:** Shivani Chauhan, Yogender Pal Khasa

**Affiliations:** Department of Microbiology, University of Delhi South Campus, New Delhi 110021, India; shivanichauhan109@gmail.com

**Keywords:** chimeric vaccines, virus-like particles, heterologous expression, bioprocess development, high-throughput technology

## Abstract

Vaccines are integral to human life to protect them from life-threatening diseases. However, conventional vaccines often suffer limitations like inefficiency, safety concerns, unavailability for non-culturable microbes, and genetic variability among pathogens. Chimeric vaccines combine multiple antigen-encoding genes of similar or different microbial strains to protect against hyper-evolving drug-resistant pathogens. The outbreaks of dreadful diseases have led researchers to develop economical chimeric vaccines that can cater to a large population in a shorter time. The process development begins with computationally aided omics-based approaches to design chimeric vaccines. Furthermore, developing these vaccines requires optimizing upstream and downstream processes for mass production at an industrial scale. Owing to the complex structures and complicated bioprocessing of evolving pathogens, various high-throughput process technologies have come up with added advantages. Recent advancements in high-throughput tools, process analytical technology (PAT), quality-by-design (QbD), design of experiments (DoE), modeling and simulations, single-use technology, and integrated continuous bioprocessing have made scalable production more convenient and economical. The paradigm shift to innovative strategies requires significant attention to deal with major health threats at the global scale. This review outlines the challenges and emerging avenues in the bioprocess development of chimeric vaccines.

## 1. Introduction

The successful vaccine development dates back to when Edward Jenner inoculated cowpox vaccinia to combat the deadly smallpox disease. Later in the 19th century, Pasteur used inactivated or attenuated microorganisms to provide successful immunity against several diseases. The successful immunization events further promoted the use of first-generation vaccines, i.e., killed, live attenuated microorganisms, or toxins, to provide adequate immunization and successful eradication of disease-causing agents [1].

Live attenuated vaccines are composed of attenuated viral or bacterial strains with immunogenic capabilities. A notable successful example is the BCG vaccine against tuberculosis, wherein Bacillus Calmette–Guerin is derived from a live attenuated strain of *Mycobacterium bovis*. The attenuation of microbial strains is accomplished by their cultivation or passaging under stressful conditions for several generations. However, live attenuated vaccines are associated with various risk factors, such as reversion to putative virulent strain and low shelf life [2]. The killed or inactivated vaccines are strains of bacteria or viruses, inactivated using harsh conditions like strong chemicals and higher temperatures. Effective killing ensures an immunogenic response with negligible reversion chances to virulence, thus providing a stable shelf life. Various shortcomings of these vaccines are higher production costs, lower immunogenic responses, and requirements of multiple booster doses [3]. Further, inactivated toxins and pathogen-derived antigens are other forms of traditional vaccines. The toxoid or inactivated toxins are prepared by chemically inactivating the antigenic toxins. Two prominent examples of prophylactic-inactivated toxins are formaldehyde-treated tetanus and diphtheria toxin [4]. Pathogen-derived vaccines constitute inactivated surface antigens, such as polysaccharides or protein-based antigens. Clinically pertinent pathogen-derived vaccines currently being used are the meningococcal vaccine (surface polysaccharides of *Neisseria meningitidis*), hepatitis B vaccine (surface antigen suspension of hepatitis B), and pneumococcal vaccine (purified surface polysaccharide antigens of *Streptococcus pneumoniae* strains) [5,6,7]. Regardless of successful immunization against various pathogens, the applicability of these traditional vaccines is constrained due to various limitations, such as more extended time frames for their development, inefficiency, safety concerns, unsuitability for non-culturable microbes, high genetic variability among microbial antigens, and the evolution of deadly pathogens.

The limitations associated with traditional vaccines have led researchers to utilize recombinant DNA technology (RDT) to develop new-generation vaccines. Modern vaccines include subunit vaccines, virus-like particles, recombinant protein-based vaccines, live vector-based vaccines, naked-DNA- and mRNA-based vaccines, peptide epitope-based vaccines, and synthetic recombinant vaccines [8]. Subunit vaccines necessitate immunogenic components of virulent strains that can stimulate antigenic response without potential exposure to the live virulent strain. Therefore, immunogenic surface and envelope protein of pathogenic strains are cloned and expressed in heterologous hosts aided by molecular biology techniques [9]. Virus-like particles (VLPs) are types of subunit vaccines that mimic the viral particle without carrying any genomic components. The structural proteins are cloned and expressed in heterologous hosts in order to be assembled as actual viral particles capable of generating an active humoral and cellular immune response [10]. Conjugate vaccines are another subunit vaccine consisting of antigenic polysaccharide components of pathogenic bacteria. These glycogenic components elicit reduced immunogenic response, thus requiring conjugation with carrier proteins. Prevnar13^®^ (Pfizer Inc., New York, USA) is a clinically relevant conjugate vaccine consisting of polysaccharides from 13 serotypes of *Streptococcus pneumoniae* conjugated with non-toxic *diphtheriae* CRM197 carrier protein [11,12]. The formulation and success of subunit vaccines depend on selecting the antigenic component, which requires the molecular-level characterization of the targeted pathogen and its associated safety concerns. Table 1 lists various commercially approved recombinant vaccines produced in different expression hosts to facilitate prophylactic treatment against various life-threatening diseases.

Another breakthrough in the current immunization program includes designing and formulating nucleic-acid-based DNA and mRNA vaccines. The naked plasmid DNA coding for immunogenic components of a pathogenic strain is directly administered intramuscularly with the help of DNA delivery systems. The recombinant plasmid DNA is intended to express antigen-encoding genes in the host cell and is targeted to stimulate both humoral and cytotoxic immune responses [27]. In comparison, RNA-based vaccines constitute mRNA or self-amplifying RNA replicons. An engineered mRNA entails an ORF (open reading frame) of the targeted immunogen, flanked by untranslated regions and a poly(A) tail sequence. These vaccines directly translate in the cytoplasm and support higher expression levels of antigenic components without being integrated into the host genome. The mRNA vaccine against the SARS-CoV-2 virus has recently proven beneficial and changed the perspective toward modern vaccination strategies [28]. Despite the advances in recombinant DNA technology, these nucleic acid-based recombinant vaccines often suffer various limitations, such as poor stability, low immunogenicity, inefficiency in vaccine delivery, and risks of genomic integration [29].

Nevertheless, modern vaccines are a proven boon to humanity in protecting against various deadly diseases. The major limitation of these vaccines is their inefficiency in providing cross-protection against multiple pathogenic strains. Immunization against hyper-evolving infectious agents has become difficult by applying conventional strain-specific recombinant vaccines. A recent example is the SARS-CoV-2 (severe acute respiratory syndrome-coronavirus-2) virus, which evolved to multiple lethal variants and continued to cause health complications. Therefore, developing a universal vaccination strategy has become a global health priority to counteract re-emerging disease-causing microbes. The goal to deliberately encounter the increasing multidrug resistance and detrimental mutations in infectious agents has led researchers to develop ‘chimeric vaccines’. The technological advancements in contemporary vaccine development would provide new avenues to combat deadly diseases.

## 2. Chimeric Vaccines: Future of Vaccinology with State-of-the-Art Technologies

Chimeric vaccines are recombinant vaccines constituting genetic or structural components of two or more microorganisms belonging to similar or different serotypes, species, or genera. A sizeable advancement in the field of vaccinology and recombinant DNA technology has helped in the development of chimeric vaccines. Conventional vaccines entail antigen-encoding gene or protein components derived from a single pathogenic strain. In contrast, the prophylactic chimeric vaccines are developed by substituting the genetic regulatory sequence or protein-encoding sequence of one microbial strain with the corresponding genes of the same or other microbial strain [30]. Chimeric vaccines leverage the concept of cross-reactivity, where shared or conserved epitopes among related pathogens are targeted to generate a protective immune response against multiple pathogens in a single formulation. Additionally, these vaccines are rationally designed to selectively incorporate putative immunodominant B-cell and T-cell epitopes for successful antigen presentation to antigen-presenting cells (APCs) and further activation of B-cell- and T-cell-specific immune response [31]. The pre-eminence of chimeric vaccines over conventional vaccines with added advantages involves:Robust immune response against multiple infectious agents;Elevated humoral response generated by B-cell lymphocyte epitopes;Elevated cytotoxic immunity elicited by T-cell lymphocyte epitopes;Good safety profile with no risk of exposure to infectious agents;Provides cross-neutralization activity in a single formulation;Effective against multi-drug resistant pathogens;Selected adjuvants provide an elevated immune response;Higher stability;Ease of storage and transportation;Rapid and reproducible large-scale production.

### 2.1. Strategies to Design Chimeric Vaccines

The strategies applicable for new-generation chimeric vaccine development include reverse vaccinology and omics-based approaches such as genomics and proteomics. Reverse vaccinology is a revolutionary in silico data-based approach to vaccine development. The strategy involves the computational rational designing of chimeric vaccines via rigorous genome screening or analysis, followed by selecting putative antigens and epitopes with immunogenic capabilities. It has been widely used to develop chimeric vaccines even against non-cultivable pathogens whose whole genome sequence is available. Moreover, developing omics-based chimeric vaccines in a shorter time eliminates the hurdles of vaccine stockpiling for re-emerging diseases [32]. Figure 1 illustrates the in silico strategies to develop different types of chimeric vaccines.

Typically, chimeric vaccine development begins with the genome and proteome screening of microbial strains to select immunogenic genes or proteins via bioinformatics studies. Computational epitope mapping strategies further allow for the selection of immunogenic linear and discontinuous B-lymphocyte epitopes, cytotoxic T-lymphocyte (CTL) epitopes, and helper T-lymphocyte (HTL) epitopes of the antigenic component. Various online immunoinformatic tools are widely employed to determine the predicted epitopes’ antigenicity, allergenicity, physiochemical properties, stability, and binding properties. The empirical selection of multiple epitopes containing major histocompatibility complex (MHC) recognition motifs effectively provides immunization with robust humoral and cytotoxic responses against several pathogenic strains in a single vaccine preparation. The development of a computationally designed chimeric vaccine is accompanied by the examination of various attributes of the vaccine construct, such as physio-chemical properties, stability, immunogenicity, tertiary structure prediction, refinement and validation of the three-dimensional (3D) structure, molecular docking studies, molecular dynamics (MD) simulations, and in silico analysis of immunogenic response [33]. In a recent study, researchers developed an integrated chimeric vaccine against sexually transmitted pathogens, such as herpes simplex virus type-2 (HSV-2), *Chlamydia trachomatis* (CT), and human papillomavirus (HPV). The bioinformatics analysis allowed the selection of potential antigens, including E7, L2 peptide of HPV, glycoprotein D of HSV-2, and outer membrane protein A of *Chlamydia*. The chimera was designed to incorporate antigenic flagellin D1/D0 Toll-like receptor 5 (TLR5) agonist, universal T-helper tetanus toxoid epitope P30, and Toll-like receptor 4 (TLR4) agonist as potential adjuvants. The study was followed by selecting epitopes of helper T-lymphocytes and cytotoxic T-lymphocytes aided with epitope mapping strategies. The computational tools, including homology modeling and molecular dynamic simulations, further validated the structure and immunogenicity of the multi-epitope chimeric vaccine [34].

Various strategies to improve the designing of chimeric vaccines require the inclusion of specific immunogenic epitopes with improved affinity to the MHC complex, epitope enhancement via genetic mutations, enhanced cytotoxic T-lymphocyte (CTL) avidity, and the application of adjuvants or macromolecular protein carriers. These strategies generally aim to improve the antigen uptake, processing, and presentation of immunogens by antigen-presenting cells [35]. In a recent in silico approach, immunoinformatic tools were employed to design a potential chimeric vaccine based on multiple pathogenesis factors of *Salmonella*. Bioinformatics tools predicted the immunogenic HTL, CTL, and B-cell epitopes of selected antigens. The in silico tools indicated the development of structurally stable chimera, constituting TLR5-binding capabilities with an anticipated potential to induce humoral and cytotoxic immunity [36].

In conclusion, the fundamental requirements for new-generation chimeric vaccine development involve identifying and selecting suitable antigenic components and precise assessment of vaccine efficacy for clinical applications. Therefore, current vaccine research is focused on developing predictive immunogenic markers, including the ability to correlate it with vaccine-derived protective immunity. Wherein, the omics-based approaches, such as genomics and proteomics, play a substantial role in investigating novel vaccine targets.

The omics-based genome mining strategy facilitates the identification of plausible antigens, which could elicit protective immunity, even applicable to non-cultivable microbes. Comparative and pan-genomics provide deeper insights into inter- and intra-species antigenic epitope diversity and distribution. Interspecies genome analysis is essential for eliminating antigens with higher similarity with human-like genes to avoid allergic reactions of antibodies raised against the immunogenic vaccine. At the same time, comparative genomics facilitates the analysis and identification of conserved genes among known infectious agents responsible for pathogenesis [37]. A comprehensive genomics study recently allowed investigators to identify immunogenic epitopes to develop a chimeric vaccine against drug-resistant *Mycobacteroides.* The genome mining of five pathogenic strains deciphered the relatedness, tandem repeats, presence of pseudogenes, antimicrobial resistance, prophage, and CRISPR/Cas genes. Pan-genomics analysis further identified the core, accessory, and unique genes in the repertoire of the *Mycobacteroid* strain’s genes. The first phase of pan-core analysis in the study indicated reductive evolution and provided a comprehensive genome analysis for constructing a potential chimeric vaccine [38].

However, the new-age chimeric vaccine development prefers proteomic approaches over genomic analysis due to the inadequate information provided by the gene sequence and mRNA. The protein expression analysis and its post-translational modifications (PTMs) allow for the examination and selection of potentially immunogenic antigens. Recent proteomics-based trends in vaccine development revolve around identifying immunogenic surface-exposed antigens, which can activate an immune response [39]. An immunoinformatics approach assisted by proteome screening has been successfully used to design multi-epitope chimeric vaccines against *Hendra henipavirus* and *Crimean-Congo hemorrhagic fever virus* [40,41]. In another study, the proteome screening of human papillomavirus (HPV) via computational tools identified highly conserved epitopes from the N-terminus of HPV-58 L2 capsid protein. The chimera designed via in silico tools constitutes two immunogenic L2 epitope sequences, N- and C-terminal regions of flagellin, short TLR4 agonist ligand peptide sequence (RS09), and two T-helper cell epitopes. The structure and immunogenicity of the chimeric vaccine were further validated via immunoinformatic tools, docking studies, and modeling simulations [42].

### 2.2. Types of Chimeric Vaccines

Chimeric vaccines developed so far are based on live attenuated viruses, VLPs, DNA, mRNA, and protein components. Live attenuated chimeric vaccines involve genetic fusion or the insertion of targeted antigen-encoding genes in the same or different live attenuated microbial strain. A notable example is the clinically approved chimeric flavivirus-based dengue virus vaccine, DENVAXIA^®^ (Sanofi Pasteur, Inc., Paris, France). The tetravalent chimeric vaccine is developed by substituting the premembrane (prM) and envelope (E) genes of 17D204 yellow fever virus (YFV) with the structural genes (prM and E) of dengue virus (DENV) strains 1, 2, 3, and 4 [43]. The development of chimeric vaccines is crucial for such viruses, constituting distinct but closely related viral serotypes capable of causing various diseases. A DENV vaccine based on insect-specific Binjari flavivirus (BinJV) has also been developed. An attempt to develop a novel immunogenic chimeric vaccine against the dengue virus was made by successfully substituting the structural prM/E genes of DENV-2 in the BinJV genome [44]. Recently, several flavivirus-based live attenuated chimeric vaccines have been under clinical trials to address the menace of West Nile, Japanese encephalitis, and Zika virus [45,46,47]. Imojev^TM^ (Sanofi Pasteur, Inc., Paris, France) is another clinically approved live attenuated chimeric vaccine, developed by replacing the structural genes of the 17D204 yellow fever virus with the prM/E genes of Japanese encephalitis SA-14-14-2 viral strain. The chimeric vaccine was highly efficient, eliminating the need for multiple doses to provide adequate immunity. Various advantages conferred by live attenuated chimeric vaccines over traditional vaccines are genetic stability, elevated immunogenicity, cross-protection, low-cost production in a shorter time frame, and minimum chances of reversion to a virulent strain [48].

The chimeric VLP (cVLP) vaccines constitute heterologous antigens displayed on the surface of VLPs, providing a broad range of immunogenic activity. The genetic or chemical modifications of native VLPs allow for the development of bivalent or multivalent chimeric vaccines, providing cross-neutralization activity. The modification strategies of VLPs include the insertion of foreign antigens in the viral structural components, surface-exposed loop, native or truncated N- or C-terminus of viral capsid proteins, transmembrane domains, cytoplasmic tails, or its fusion with GPI (glycosylphosphatidylinositol) anchors [49]. The DNA-sequence-encoding self-assembling VLP-forming polypeptides is fused with the gene-encoding target epitope or antigen to yield chimeric VLPs carrying the immunogen in high density and stable conformation. Another approach involves in vitro chemical conjugation of pre-assembled VLPs and foreign antigens via covalent or non-covalent interactions [50]. Chimeric VLPs serve numerous advantages over other vaccine candidates. The chimeras mimic wild-type viruses in shape and geometry, elicit an elevated humoral and cytotoxic immune response, eliminate risks of viral exposure, encode T-helper cell epitopes for antigen presentation, and allow for the secretion of immunomodulatory cytokines. Chimeric VLPs could also be engineered to carry exogenous molecules and adjuvants, making them a perfect choice for chimeric vaccine development platforms [51]. Mosquirix^TM^ (GlaxoSmithKline, Brentford, UK) RTS, S/AS01 is a commercially available chimeric-VLP-based vaccine developed against malaria. The vaccine candidate ‘RTS, S/AS01’ was designed to consist of the highly conserved four-amino-acid-tandem repeats of the central repeat region (R) and C-terminal region of *P. falciparum* NF54 circumsporozoite protein (CSP), consisting of immunodominant T-cell epitopes (T). The genetically engineered ‘RT’ fragment was fused to hepatitis B surface antigen (HBsAg) at the N-terminus to generate self-assembling chimeric VLPs, termed as RTS. In S/AS01, the S represents the unfused independently assembled HBsAg-based VLPs, and AS01 represents the adjuvant added to vaccine formulation to improve immunogenicity. The vaccine was designed to generate robust anti-CSP antibodies and a cytotoxic T-cell immune response. This commercial chimeric VLP-based vaccine exhibited promising therapeutic results in clinical trials and emerged as the first chimeric vaccine approved for immunization against malaria [52].

Chimeric vaccines also constitute multi-epitope-based vaccines, which combine the immunodominant epitopes of desired antigens. These vaccines are targeted to generate robust and extended B-cell- and T-cell-specific immune responses, eliminating the need for multiple vaccine doses. A recent study illustrated designing a novel chimeric vaccine against SARS-CoV-2 and its associated secondary infections. The secondary co-infections caused by commensal bacterial strains, such as *Streptococcus pneumoniae*, *Haemophilus influenzae*, and *Mycobacterium tuberculosis*, elevate the severity of SARS-CoV-2 infection by manifolds, thus rendering the treatment more challenging. Hence, the B-cell, HTL, and CTL non-toxic immunogenic epitopes of selected strain-specific antigens were screened and designed via molecular simulations to develop a potent chimera against SARS-CoV-2 and the associated secondary pathogens. The in silico evaluation of multi-epitope chimeric vaccine demonstrated the elicitation of high-antibody titers [53]. Another study demonstrated chimeric protein ‘ChimT’ development against *Leishmania infantum*, the causative agent of visceral leishmaniasis. The specific CD4^+^ and CD8^+^ T-cell epitopes of immunodominant *Leishmania* amastigote proteins were selected via bioinformatics tools to develop the chimeric protein. The novel chimeric vaccine candidate constituting the ChimT protein and potential adjuvants generated a strong cytotoxic immune response accompanied by cytokine production in the mice models [54].

Advances in molecular techniques have provided the opportunity to develop nucleic-acid-based chimeric vaccines. DNA-based chimeric vaccines are engineered plasmid DNA constituting antigen-encoding genes from two or more microbial strains. These vaccines are versatile, stable, easier to construct, generate higher antibody and cytotoxic immune responses, and allow for economical scale-up and mass production. Moreover, technological advancements in DNA delivery systems, such as sophisticated electroporators, liposomes, and nanoparticles, have made safe and efficient chimeric DNA-based vaccination. Chimeric DNA-based vaccines also find immense application in cancer immunotherapy, where the foreign immunogen of a chimeric vaccine can restrict tumor growth by providing a cross-reactive T-cell immune response against self-tolerated tumor-associated antigens [55]. A DNA-based chimeric vaccine combining the virulence genes of *Vibrio cholerae* OmpW (outer membrane protein), CtxB (cholera toxin subunit B), and TcpA (toxin co-regulated pilus), has been successfully developed. The immunization with DNA-based chimera generated elevated IgG titers, increased survival rates, and inhibited lethal bacterial infection in mice models [56]. A recent study demonstrated the efficacy of receptor binding domain (RBD)-based chimeric DNA vaccine against SARS-CoV-2. The chimeric plasmid DNA was constructed by inserting the mutated preS1 region of the hepatitis B virus at the N-terminus of SARS-CoV-2 receptor binding domain. The in vitro and in vivo experiments indicated enhanced inflammatory cytokine production, elevated cytotoxic immune response, and higher titers of RBD-specific IgG and IgA antibodies in mice immunized with chimeric DNA. Further, in vitro studies suggested no antibody-mediated antibody-dependent enhancement (ADE)-related immunopathology of SARS-CoV-2 in the vaccinated mice model [57]. Chimeric mRNA-based vaccine constitutes exons derived from different microbial strains capable of generating a robust cross-protective immune response. Moreover, mRNA-based chimeric vaccines eliminate the need for genomic integration and pose no risk or safety concerns. These advantages led researchers to develop a chimeric prophylactic mRNA-based vaccine effective against the omicron and delta strains of SARS-CoV-2. A chimeric vaccine based on mRNA was developed by introducing the receptor-binding domain of the delta strain in the spike (S) antigen-based mRNA of the omicron strain. The engineered mRNA elicited neutralizing antibodies against both strains, indicating a wide-ranging potential of the chimeric vaccine in various immunization programs globally [58].

## 3. Bioprocess Development of Chimeric Vaccines: Challenges and Opportunities

The global emergency due to COVID-19 (coronavirus disease 2019) has led to over 6.9 million deaths worldwide as of May 2023 [59]. The outbreaks require a lot of infrastructure and facilities to produce medical aid on a larger scale. Therefore, developing prophylactic chimeric vaccines and therapeutics against hyper-evolving pathogens has become a priority. In this regard, recombinant DNA technology and bioprocess engineering have been demonstrated as an advantage for the pharmaceutical industry in developing chimeric vaccines comprising immunogens against multiple pathogens.

The process development of chimeric vaccines has been divided into two main parts, i.e., upstream and downstream bioprocessing. The upstream development involves selecting and cloning synthetic genes in expression vectors, followed by transformation and protein expression in heterologous hosts. In contrast, the downstream processing consists of the purification and formulation of the vaccine candidates, which is essential to obtain regulatory approvals [60]. A schematic representation of the different stages involved in chimeric vaccine development has been given in Figure 2.

The bioprocess development of difficult-to-express chimeric vaccines involves various developmental stages. Due to genetic variability and size, manufacturing these vaccines is challenging and requires more robust bioprocess techniques. Various challenges faced during chimeric vaccine development involve selecting suitable heterologous hosts, cultivation conditions, and effective purification strategies. The prime concern involved in the bioprocess development of chimeric vaccines is the assembly of antigens from different strains of pathogens. Various bottlenecks associated with the assembly and production of heterologous immunogens are product instability, poor solubility, modified physiochemical properties, and altered glycosylation profiles. The introduction of foreign antigens renders protein to adopt different quaternary structures. These process-related issues often produce misfolded or inactive chimeric immunogens with poor homogeneity and immunogenicity [61]. However, selecting and optimizing critical quality attributes (CQAs) are essential to overcome the challenges mentioned earlier. The advanced technologies and automation in upstream and downstream processes provide innovative solutions and opportunities in vaccine development maneuvers. Therefore, various challenges and future trends in the bioprocess development of chimeric vaccines have been comprehensively deliberated in the following sections.

## 4. Expression Platforms and Upstream Process Development of Chimeric Vaccines

The decisive factors governing the upstream process development of chimeric vaccines are vaccine demand, type of vaccine, expression requirements of immunogenic antigen, host cell characteristics, growth requirements, and vaccine stability. The upstream process development begins with the selection of the expression vector and host organism, which are imperative to achieve high product yields. Various expression platforms have been explored to produce these vaccines, such as bacteria, yeasts, insect cell lines, mammalian cell lines, transgenic plants, and microalgae. Various other factors that combinatorially affect the upstream process development of chimeric vaccines are the selection and optimization of media components, selection of cultivation mode, selection of bioreactor type, optimization of cultivation conditions, and scale-up of the fermentation process. Bioprocess development strategies are initially tested at a laboratory scale to optimize various cultivation conditions to favor large-scale production at the manufacturing facility [62].

### 4.1. Bacterial Expression System

*E. coli* is the most preferred prokaryotic expression host for producing therapeutic products and vaccines. It offers numerous advantages over other expression platforms due to its well-established genetics and growth kinetics, higher productivity rates, ease of genetic modifications, and economical production. However, it also has various limitations, including lack of post-translational modifications (PTMs), the formation of aggregates or inclusion bodies, issue of codon bias, antigenic toxicity, and contamination by endotoxins or lipopolysaccharides [63]. Nevertheless, recent advances in strain improvement have opened various avenues to combat these disadvantages. It has been extensively investigated to produce chimeric vaccines against evolving pathogens. *E. coli*-expressed chimeric multivalent (triple-type) vaccines against human papillomavirus (HPV) have been well characterized and resulted in promising translatability and qualities [64,65]. Middelberg et al. [66] utilized *E. coli* host as an effective platform to produce prophylactic chimeric VLPs based on murine polyomavirus VP1 capsomeres. The M2e peptide from the influenza H1N1 virus and the J8 peptide from Group A *Streptococcus* (GAS) protein were presented on the C-terminal-truncated mutant of murine polyomavirus (MuPyV) VP1 structural protein. The successful preparation of MuPyV-based chimeric self-adjuvating VLPs in *E. coli* Rosetta (DE3) pLysS cells demonstrated rapid and cost-effective manufacturing of chimeric vaccines. The bioactivity of J8-VP1 chimeric VLPs was examined, which indicated higher J8-specific serum IgG antibodies with opsonizing capabilities in mice model. The high-level expression of heterologous immunogenic antigens in *E. coli* often renders proteins to form inclusion bodies. Piubelli et al. [67] developed different strategies to obtain a soluble expression and production of a potential *M. tuberculosis* (MTB) vaccine in *E. coli* with the help of an N-terminal thioredoxin solubility tag. The codon-optimized chimeric TB10.4-Ag85 fusion proteins’ gene was rationally designed, combining immunogenic Ag85B mycolyl transferase and TB10.4 belonging to the esat-6 gene family. The study highlighted the scalable soluble expression of a chimeric antigen with elevated volumetric and specific productivity at the bioreactor level. Using the *E. coli* expression system, the large-scale process development for chimeric repeats VK210 and VK247 of *P. vivax* circumsporozoite antigen was optimized in a cGMP (current Good Manufacturing Practices) facility at 300 L bioreactor facility with high product recovery [68].

### 4.2. Yeast Expression System

Yeasts are eukaryotic heterologous hosts that grow on a simple growth medium, offering easy genetic manipulations and scale-up. These unicellular microorganisms contain sub-cellular machinery, which allows for post-translational modifications. Yeasts-based vaccines for clinical application include whole-cell recombinant yeast or purified recombinant antigens expressed in yeast cells [69]. Different yeast expression platform explored for chimeric vaccine production involves conventional yeasts, such as *Saccharomyces cerevisiae*, *Komagataella phaffii* (formerly known as *Pichia pastoris*), and *Schizosaccharomyces pombe*, and other non-conventional yeasts, such as *Hansenula polymorpha*, *Yarrowia lipolytica*, *Kluyveromyces lactis*, and *Arxula adeninivorans* [70,71].

Various advantages of using *Saccharomyces* as an expression host involve tolerance to a wide range of pH and osmotic pressure, optimum expression level, proper folding, post-translational modifications, easy scale-up, and economical production. It was successfully used for the commercial production of various human vaccines against hepatitis B virus (Recombivax HB^®^, Merck & Co., Inc., NJ, USA; Engerix^®^ B, GlaxoSmithKline, Brentford, UK; HBVaxPRO^®,^ Sanofi Pasteur MSD, Lyon, France; and Fendrix^®^, GlaxoSmithKline, Brentford, UK), human papillomavirus (Gardasil^®^, Gardasil-9^®^_,_ Merck & Co., Inc., NJ, USA), and malaria (Mosquirix^TM,^ GlaxoSmithKline, Brentford, UK) [72]. Powilleit et al. [73] demonstrated the development of a novel *S. cerevisiae*-based platform to produce chimeric VLPs. The fusion of the C-terminal Gag protein of totivirus and the N-terminally truncated pp65 variant of human cytomegalovirus (HCMV) resulted in the self-assembly of immunogenic chimeric VLPs in vivo. These VLPs can express cytotoxic protein and prevent proteolytic degradation of the cargo protein in the cytosol of the host organism. However, one of the significant drawbacks of using *Saccharomyces* involves the hyper-mannosylation of recombinant therapeutics, which is immunogenic and causes a short serum half-life. *Saccharomyces cerevisiae* has also been investigated for the production of a chimeric vaccine targeting enterovirus 71 (EV-A71) and coxsackievirus A16 (CVA16), causing hand-foot-and-mouth disease. The potential chimera was designed by replacing the neutralizing epitope (SP70) of the VP1 structural protein of EV-A71 with the corresponding epitope of CVA16 to develop ChiEV-A71 VLPs. The *Saccharomyces*-expressed chimeric VLPs were structurally stable and elicited robust humoral and cytotoxic immune responses against both viruses in mice models [74].

*Komagataella phaffii*, formerly *Pichia pastoris*, has emerged as a “Biotech yeast” to express various recombinant vaccines. The added advantages of using *K. phaffii* over *Saccharomyces* involve easy maintenance and scalability, high-cell-density cultivation, high-level protein expression, the multicopy genomic integration of foreign genes, ease of genetic engineering, the efficient assimilation and catabolism of 1-carbon compound, gene regulation via tightly controlled methanol inducible AOX1 (alcohol oxidase 1) promoter, reduced hyper-glycosylation, the high-level extracellular secretion of proteins, and economical production [75]. *Pichia pastoris* has been widely used in developing chimeric vaccines against emerging viruses, offering an effective and safe vaccination approach. The development of various chimeric VLPs and recombinant antigens using the *Pichia* expression platform has been summarized in Table 2.

Chimeric vaccine production in *P. pastoris* is often facilitated by the fed-batch mode of cultivation, which involves the continuous feeding of growth-limiting nutrients during the cultivation. It prevents product inhibition and supports high-cell-density growth with ameliorated product yields [93]. The fed-batch mode of cultivation has also been utilized to produce bioactive antitumor ATF-SAP fusion chimera in a 2 L stirred tank bioreactor. The difficult-to-express chimera contained an amino-terminal fragment (ATF) of urokinase and plant-derived saporin toxin (SAP). The optimization of process parameters, such as growth media, temperature, pH, DO (dissolved oxygen), feeding strategy, and growth rate, led to improved protein production, i.e., ~12.5 mg/L in the *P. pastoris* GS115 strain. The therapeutic potential was confirmed by the cytotoxic activity of the chimera against uPAR (urokinase-type plasminogen activator receptor)-overexpressing U937 leukemia cell line [94]. In one of the studies, the investigators aimed to develop a recombinant chimeric vaccine against human papillomavirus (HPV) and human immunodeficiency virus (HIV) using the *Pichia* expression platform. The L1 capsid protein of HPV-16 and the P18-I10 CTL epitope (V3 loop of gp120) of HIV-1 were selected to design the L1P18 chimeric gene. The codon-optimized gene was cloned in the pGAPZ B vector and constitutively expressed in the *P. pastoris* X-33 strain. The intracellularly expressed chimeric L1P18 protein was purified via size exclusion chromatography, ultracentrifugation, and ultrafiltration with 96% purity and a recovery yield of 9.23%. The approach highlights bivalent vaccines’ easy and reliable production against major global pathogens using an alternative yeast-based expression platform [95]. In another study, researchers developed a multi-component malarial vaccine consisting of distinct parasite ligands of *Plasmodium falciparum*, namely, PfAMA-1 (domain III of apical membrane ag-1), PfMSP1 (merozoite surface protein), and PfEBA-175 (binding domain for glycophorin A on erythrocytes). The initial studies developed a chimera constituting a C-terminal fragment of PfAMA-1 fused with PfMSP1. The generated fusion construct PfCP-2.9 was successfully expressed as an immunogenic protein in *P. pastoris* GS115 with a product yield of 2600 mg/L [96]. The group continued to develop the F2 domain of the PfEBA-175 protein in *P. pastoris* GS115 with a final yield of 300 mg/L. The combinatorial dosage of PfEBA-175 and PfCP-2.9 chimeric protein successfully produced an immunogenic response in the mice, rabbit, and rhesus monkey models. The antisera isolated from the model organisms inhibited the in vitro growth of merozoites, thus suggesting the application of these antigenic proteins in developing a multivalent vaccine against malaria [97]. To develop novel vaccine candidates against the influenza virus, tandem core technology was used, in which the highly hydrophobic long alpha helix (LAH) domain of the H3N2 influenza strain was incorporated in the chimeric VLPs derived from hepatitis B virus core protein [98].

The worldwide health emergency due to COVID-19 led researchers to identify new vaccine candidates, wherein *P. pastoris* served as a potential heterologous expression host. One of the widely studied antigens for SARS-CoV-2 vaccine development is the spike protein’s receptor-binding domain (RBD). The RBD aids the virus’s entry into the host cell via the ACE2 (angiotensin-converting enzyme 2) receptor protein. The engineered RBD, i.e., C-RBD-H6, was successfully produced in a 50 L bioreactor using fed-batch fermentation strategies in *Pichia pastoris* X-33. The glycosylated protein was purified via affinity chromatography to yield highly immunogenic recombinant RBD to a level of 30–40 mg/L with >98% purity [99].

*Hansenula polymorpha* is a non-conventional methylotrophic yeast engineered and developed to produce various modern vaccines. Like the above-mentioned conventional yeasts, it confers various advantages as an expression host, such as tolerance to high-temperature range (30–50 °C), ability to utilize methanol, the ease of genetic engineering, heterologous gene regulation via strong methanol-inducible promoters, improved expression yields, and also the circumvention of the hyper-mannosylation of recombinant proteins. To date, clinically approved hepatitis B vaccines have been produced using *H. polymorpha*, including HepavaxGene^®^ (GreenCross Vaccine Corporation, Seoul, Korea), GeneVac B^®^ (Serum Institute of India, Pune, India), Heplisav-B^®^ (Dynavax Technologies Corporation, Emeryville, USA), and Biovac-B^®^ (Wockhardt Ltd., Mumbai, India) [100]. The development of a vaccine based on hepatitis B surface antigen in *H. polymorpha* was first initiated in 1991 [101]. Further, the C-terminal-truncated PreS2 (120–145 amino acid) in fusion with a complete hepatitis B surface antigen sequence yielded up to 250 mg/L of recombinant product in *H. polymorpha* DL-1 strain under strong inducible MOX (methanol oxidase) promoter [102]. *H. polymorpha* has also been used to produce and develop L1 and L2 capsid protein-based human papillomavirus (HPV) vaccines. The fed-batch fermentation of *H. polymorpha* produced higher yields of HPV-16 L1-L2 chimeric antigen. The chimera was developed by substituting the h4 helix of HPV-16 L1 capsid protein with the corresponding L2 peptide. The recombinant gene was cloned in pHIPX4–HNBESX plasmid, followed by expression studies on *H. polymorpha*. The study suggested that methanol feeding via the DO stat method in fed-batch fermentation led to a considerable increase in biomass and protein production up to 132.10 mg/L [82]. Attempts were made to increase the expression level of chimeric HPV-16 protein via peroxisomal targeting using signal sequence. However, the targeted expression in the peroxisomes of methylotrophic yeast negatively impacted the chimeric protein yields [103]. A vaccine candidate based on HPV type-6/11 has also been developed, from which the yield of HPV-6 L1 and HPV-11 L1 proteins were 408 mg/L and 537 mg/L, respectively. Further, these proteins were subjected to disassembly and reassembly treatment to yield stable and dispersed VLPs, which elicited neutralization antibodies in mice and monkey models [104].

### 4.3. Insect Cell Expression System

The baculovirus–insect cell expression host has been exploited as a workhorse of recombinant protein production at an industrial scale. Various advantages attributed to this expression platform have led to the development of commercial prophylactic vaccines against evolving global pathogens [105]. Table 3 lists commercially approved vaccines developed in the insect cell expression system for human and veterinary use. Baculoviruses are enveloped ds DNA viruses with large genome sizes, allowing them to incorporate multiple foreign DNA sequences. The two widely studied baculoviruses are *Autographa californica* multiple-capsid nuclear polyhedrosis virus (AcMNPV) and *Bombyx mori* nucleopolyhedrovirus (BmNPV). However, AcMNPV is a model system with many insect hosts to produce commercial recombinant proteins [106]. Standard insect cell lines used for the propagation of recombinant baculoviruses are derived from ovarian cells of *Spodoptera frugiperda* (Sf-21, Sf-9, and expresSF+^®^) and cabbage looper (*Trichoplusia ni*, High Five^TM^) [107]. The recombinant vaccine development via the baculovirus expression vector system (BEVS) starts by generating a working virus bank (WVB), established by passaging recombinant BEVS during multiple rounds of infections in insect cells. The stably maintained high titer of the functional virus bank is then used to infect actively growing insect cells in culture flasks or bioreactors to initiate upstream production [108]. The advantages of working with BEVS in recombinant vaccine production involve easy genetic modification, post-translational modifications, the ability to correctly fold immunogenic bioactive proteins, no risk of live pathogens, the certified regulations of production, the inherent safety of baculoviruses, rapid process development, easy scale-up, and cost-effective production at the commercial level [109].

Apart from the commercially available recombinant vaccines, several vaccines are under different phases of clinical trials. A recent study suggested the development of chimeric SARS-CoV-2 VLPs via optimization of the *spike* (*S*) gene. The mutated codon-optimized spike protein gene (*mS* gene) was cloned upstream of the H5N1 *matrix 1* (*M*) gene of the influenza virus in pFastBac™ dual donor plasmid to study the shake-flask-expression of *mSM* chimeric gene in Sf-9 cells. Compared to wild-type spike (S) protein, the VLPs developed from the codon-optimized chimeric gene significantly raised the SARS-CoV-2-specific neutralizing antibodies and cellular immunity in mice models [122]. The cost-effective production of chimeric human immunodeficiency virus (HIV)-based VLPs has been successfully obtained by optimizing the BEVS system. Chimeric HIV-1 VLPs were developed by fusing Gag polypeptide precursor sequence with non-structural proteins, including reverse transcriptase (RT) and TatNef (TN). The results demonstrated that optimizing the insect cell line and infection time maximized the yields of chimeric HIV-1 GagRT and GagTN VLPs in the *Trichoplusia ni* PRO^TM^ insect cell line [123]. The baculovirus expression platform has also been evaluated to produce chimeric Sudan VLPs for the development of a prophylactic vaccine against the deadly Ebola virus. The full-length gene-encoding structural matrix protein (VP40) and glycoprotein (GP) of Sudan virus were cloned in the pFastBacDual vector to yield recombinant bacmids, i.e., pFastBacDual-VP40-VP40 and pFastBacDual-GP-GP, respectively. The transfection and co-expression of recombinant bacmids in Sf-9 cells led to the production of immunogenic Sudan VLPs, which elicited the production of neutralizing antibodies and Th1 (T helper 1)- and Th2 (T helper 2)-type cytokine-secreting immune cells in mice models [124]. Wang et al. [125] have demonstrated the potential of the baculovirus expression system in the elevated production of human papillomavirus (HPV) L1 capsid protein-based VLPs. HPV-58 L1 protein-encoding gene was mutated to generate modifications at the N- and C-terminus. The mutant was expressed in Sf-9 cells, contributing 2.3-fold higher protein production. The truncated protein was subjected to simplified two-step purification to yield 60 mg/L of the product. The purified protein, with 99% purity, successfully elicited virus-specific neutralizing antibodies, thus suggesting a cost-effective method for HPV-based chimeric VLP production using BEVS at an industrial scale. In a similar study, researchers demonstrated the high-yield production of L1 major capsid protein-based HPV L1 and chimeric L1-L2 VLPs via the substitution mutations of basic amino acids at C-terminus.. The putative mutations enhanced the protein expression of chimeric VLP (58L1-16L2) up to 3.4 fold, eliciting strong immunogenicity and stability [126]. The baculovirus expression system has also been investigated to generate a highly immunogenic chimeric vaccine candidate against the influenza virus. Chimeric VLPs displayed the H1 stem of influenza virus immunogenic hemagglutinin protein and the C-terminally-truncated DnaK protein of *E. coli*. In mice models, the Sf-9-expressed chimeric VLPs exerted robust in vivo immune protection against lethal influenza virus infection [127]. Another study demonstrated the successful production of prophylactic chimeric VLPs against the rabies virus. Recombinant baculoviruses were developed for the putative expression of glycoprotein and matrix protein of rabies virus, fused with the membrane-anchored GM-CSF (granulocyte macrophage colony-stimulating factor), wherein the GM-CSF potentially worked as an adjuvant. Compared to the standard available vaccine, the chimeric VLPs elevated the recruitment of immune cells and rabies-virus-specific neutralization antibodies in mice model [128].

### 4.4. Mammalian Cell Expression System

The mammalian cell expression platform contributes significantly to the commercial production of recombinant proteins and prophylactic vaccines. A few viral vectors widely used for mammalian cell expression are lentiviral vectors, non-viral adeno-associated virus systems, and Semliki Forest viral vectors. Recombinant mammalian cell lines are developed via the successful transfection and integration of foreign genes in the transcriptionally active region of the host cell genome [129]. The mammalian cell lines routinely used for vaccine production are Chinese hamster ovary (CHO), baby hamster kidney (BHK), Vero cell line (African monkey kidney epithelial), murine myeloma cells (NS0 and Sp2/0), human embryonic kidney 293 (HEK293), human fibrosarcoma (HT-1080), human-derived PER.C6, CAP (CEVEC’s amniocyte production), and human hepatoma (HuH-7) cell line. However, the predominantly used cell lines for vaccine production are CHO and HEK293 [130]. The selection and optimization of media components are critical for the process development of chimeric vaccines. Media components essentially provide all the nutrients required for the cultivation of the host organism, which eventually affects the production yields of chimeric vaccine. A notable example is the development of a serum-free Vero cell line culture process to produce a chimeric virus vaccine targeted against parainfluenza virus type 3 and respiratory syncytial virus. The supplementation of chemically defined lipid concentrate in the pre-infection growth medium and the reduction in post-infection cultivation temperature demonstrated a ~100-fold increase in virus titers [131]. Lorenzo et al. [132] demonstrated the effect of four commercially available culture media on producing chimeric subunit vaccine E2-CD154 against classical swine flu fever virus in a 10 L bioreactor using HEK293 mammalian cell line. The study suggested that post-translational glycosylation and immunogenic response in mice models were independent of culture conditions. However, two commercial media, namely SFM4HEK293 and CDM4HEK293, were selected due to their ability to support high specific growth of the HEK293 cell line. In a recent study, Fontana et al. [133] demonstrated the production of HIV-1 Gag-based chimeric VLPs against foot-and-mouth disease virus (FMDV) in the HEK293 cell line. The chimera was designed by inserting antigenic G-H loop of FMDV capsid protein VP1 (GH) in the rabies glycoprotein (RVG). The co-expression of fusion protein (GH-RVG) and HIV-1 Gag capsid protein in HEK293 cells yielded immunogenic chimeric VLPs. Further, bioprocess optimization strategies enhanced the yields of chimeric VLPs by 5.5-fold, indicating a convenient scale-up of novel chimeric FMDV vaccine candidate. Recently, a stable HEK293T cell line has also been established in reduced-serum media to express chimeric VLP-based SARS-CoV-2 vaccine. Immunogenic chimeric VLPs were developed consisting of spike (S) protein of SARS-CoV-2 with truncated polybasic furin cleavage motif fused with the cytoplasmic tail and transmembrane domain of influenza A hemagglutinin and M1 matrix protein. A lentiviral vector was utilized for cloning the M1 matrix and chimeric S fusion genes separately, followed by the HEK293T cell line transduction. The M1 matrix protein assisted in the continuous release of budding VLPs from the cell host, thus escaping the need for multiple transfections. In addition, these enveloped chimeric VLPs were found to be immunogenic in mice models and generated S-specific neutralizing antibodies [134].

### 4.5. Other Expression Platforms

The increased incidents of global outbreaks have led to the development of alternative platforms for chimeric vaccine production. Transgenic plants have been successfully used to express recombinant therapeutics in large quantities. Additionally, the vaccines developed in plant-based expression systems are safe and free from any mammalian-borne pathogens, thus eliminating the safety issues associated with available conventional vaccines. Plant molecular farming (PMP) has opened the avenues to produce complex chimeric vaccines, which are easily scalable and affordable [135]. Several host plants that have been used as potential expression platforms are tobacco (*Nicotiana benthamiana*, *Nicotiana tabacum*), *Arabidopsis thaliana*, rice (*Oryza sativa*), maize (*Zea mays*), potato (*Solanum tuberosum*), tomato (*Lycopersicon esculentum*), and lettuce (*Lactuca sativa*) [136]. Recently, the tobacco expression platform was used to develop a commercially available Covifenz^®^ (Medicago Inc., Quebec, Canada and GlaxoSmithKline, Brentford, UK) vaccine against COVID-19 [137]. Dobrica et al. [138] demonstrated the production of a novel, cost-effective chimeric antigen against the hepatitis B virus in *Nicotiana benthamiana.* The chimera was developed by inserting the 21–47 amino acids sequence derived from preS1 large surface protein (L) into an external antigenic loop of small surface protein (S). The chimeric antigen (S/preS1^21–47^) was transiently expressed in tobacco leaves and successfully elicited both humoral and cell-mediated immunity in mice model. In another study, the high-yield production of a chimeric human papillomavirus (HPV) 16 vaccine candidate was achieved. The study involved developing a fusion protein consisting of a modified E7 protein of HPV-16 and cell-penetrating bacterial anti-lipopolysaccharide factor LALF_(31–52)_. Higher protein yields were accomplished via the self-replicating expression vector with increased TSP (total soluble protein) accumulation up to ~27 fold in the chloroplast [139]. The researchers also developed a chimeric antigen consisting extracellular domain of cytotoxic T-lymphocyte associated antigen-4 (CTLA-4) and *E. coli*-expressed B subunit of heat-labile enterotoxin. The resultant chimera was transiently expressed in *N. benthamiana*, yielding up to 1.29 μg/g fresh leaves. The immunogenicity of LTB-CTLA4 was tested in a mice model, where the chimeric antigen successfully elicited IgG-specific antibodies against the chimeric construct [140]. Recently, the truncated coat protein of the hepatitis E virus (HEV) was used as a carrier to generate M2e peptide-based influenza A and receptor binding domain (RBD)-based SARS-CoV-2 chimeric VLPs in *Nicotiana benthamiana*. The fusion constructs HEV/M2e and HEV/RBD were successfully expressed and purified with final yields of 200 μg/g and 20 μg/g fresh leaf tissue, respectively [141].

After the development of transgenic plants as an expression platform, various microalgae systems, such as *Chlamydomonas reinhardtii*, *Phaeodactylum tricornutum*, *Dunaliella salina*, *Chlorella vulgaris*, and *Schizochytrium* sp., were investigated as expression platforms. These microalgae confer various advantages like easy maintenance, high growth rates, ease of genetic manipulation, low nutritional requirements, post-translational modifications, and correct folding of complex proteins. The vaccine candidates developed using these hosts are safe and free from endogenous toxins and animal-borne pathogens [142]. One of the widely investigated algae for expression studies is *C. reinhardtii.* The FDA (U.S. Food and Drug Administration) has approved it as Generally Regarded as Safe (GRAS). Both nuclear- and chloroplast-based expression has been investigated in these mini-cultivators [143]. Recently, a chimeric multi-epitope algae-based vaccine entailing potential antitumor activity has been developed. The chimeric construct was designed to combine B-cell epitopes of tumor-associated antigens, including human epidermal growth factor receptor-2, mucin-like glycorprotein 1, Wilms’s tumor antigen, and mammaglobin. The chimeric gene construct was cloned with the immunogenic B subunit of *E. coli* thermolabile enterotoxin (LTB) and expressed in *Schizochytrium* sp. to yield 637 µg/g fresh weight. The immunogenicity of the protein was examined in mice model, which indicated elevated IgG titers against tumor lysate [144]. The development of a whole-algal-cell-based oral vaccine derived from the β subunit of cholera toxin (CtxB) and surface protein of *Plasmodium falciparum* (Pfs25) was also established. The chloroplast of *C. reinhardtii* was engineered to express structurally complex chimeric fusion protein CtxB-Pfs25 into its soluble form. The functionality of chimeric protein was tested in a mice model, wherein the oral vaccination elicited secretory IgA antibodies against both CtxB and Pfs25 proteins [145]. The B subunit of cholera toxin has also been used to develop a prophylactic chimeric vaccine for the treatment of atherosclerosis. Successful expression of chimeric fusion containing B subunit of cholera toxin and p210 epitope of apolipoprotein ApoB100 in *C. reinhardtii* yielded antigenic active protein up to 60 μg/g of fresh weight. The chimera construct was immunogenic, as indicated by the elevated anti-p210 serum antibodies in BALB/c mice model [146]. The algae-based expression platform has also been evaluated for producing an oral vaccine against the Zika virus (ZIKV). The chimeric protein was developed in *Schizochytrium* sp., consisting of envelope protein epitopes of the Zika virus and carriers derived from the B subunit of *E. coli* thermolabile enterotoxin. The algal expression host utilized the ‘Algevir system’ to yield protein up to 365 μg/g of fresh weight. At the same time, the oral immunization of the ZIKV chimera in mice model resulted in a higher amount of antigen-specific neutralizing antibodies [147]. Several reports highlight the production of chimeric vaccine candidates in transgenic plants and microalgae; however, these systems are not yet established for the industrial-scale production of chimeric vaccines. Nonetheless, efforts are constantly being made to improve these expression platform’s upstream and downstream processes.

## 5. Emergent Technologies in Upstream Process Development

The continuous cultivation-based perfusion culture mode is an emerging technique in chimeric vaccine production, which supports a steady growth of cells in the log phase in stirred tank bioreactors. The perfusion setups are assisted by cell retention devices, including tangential flow filtration (TFF), alternating tangential flow (ATF), and acoustic settler, which allows for the continuous operation of stable, viable cell lines and continuous product removal. It confers various advantages in chimeric vaccine production over conventional technologies, such as high-cell-density cultivation, stable cell line propagation, the removal of toxic by-products, prolonged cultivation hours, the maintenance of product quality, the ability to integrate downstream processing, and low capital cost [148]. The perfusion cultures are highly advantageous for fusogenic oncovirus-based chimeric vaccines that form large multinucleated syncytia and result in low virus yields in anchorage-dependent cell lines. A recent study demonstrated the perfusion-culture-based high-yield production of oncolytic chimeric virus vector rVSV-NDV, wherein the endogenous glycoprotein of vesicular stomatitis virus (VSV) has been substituted with the envelope proteins of Newcastle disease virus (NDV). Culturing chimeric oncolytic virus assisted by an acoustic settler resulted in ˃97% cell retention efficiency, elevated virus titers, and increased volumetric virus productivity by 15–30 fold compared to batch processes [149]. In another study, researchers demonstrated the utility of continuous stirred tank perfusion bioreactors for stable constitutive expression of Zika-virus-based VLPs. The codon-optimized premembrane (prM) gene in fusion with wild-type and chimeric mutants of envelope (E) genes was transfected in the HEK293SF-3F6 cell line. The VLP production was successfully scaled up to stirred tank bioreactor with ~4-fold improved yields using a robust cell retention system of alternating tangential filtration (ATF). The results suggested that maintaining the cell-specific perfusion rates in the continuous bioreactors results in higher product yields [150].

Advanced high-throughput devices have considerably improved chimeric vaccine upstream screening and optimization processes. Notable examples of high-throughput devices are multi-well plates, mini-bioreactors, miniature shake vessels/wells, microplate-based mini-bioreactors, and stirred mini-tank bioreactors. The devices are designed to develop automated workstation, robotic feeding and sampling, optical sensors, simultaneous cultivation runs, online assessment capabilities, and automated centralized data storage [151]. In a recent study, a high-throughput cell culture reactor, i.e., Ambr15, was implemented to scale-up the macrocarrier-aided chimeric SARS-CoV-2 vaccine production in Vero cells. Ambr15 is a sophisticated robust automated high-throughput screening device with 24 individually controlled single-use stirred tank reactors. The cultivation processes, such as cell density, multiplicity of infection (MOI), media addition, media exchange, and virus production temperature, were optimized to yield maximum viral titers [152].

Recent trends in single-use cultivation technologies have opened the scope for cost-effective mass production of chimeric vaccine candidates. The technology involves the application of economical and flexible single-use tools, such as single-use probes, sensors, bioreactors, and fluidic components [153]. Recently, a serum-free microcarrier-based single-use closed bioreactor system has been utilized to produce a chimeric vaccine candidate against SARS-CoV-2 at a large scale. The chimeric vaccine candidate was developed by substituting the vesicular stomatitis virus (VSV) glycoprotein with the spike (S) protein of SARS-CoV-2. Process optimization studies were initially carried out in a 3 L BioFlo320 glass bioreactor using the design of experiment (DoE) approach, where optimal cultivation temperature of 34 °C and pH-7 improved virus productivity by ~1 log. Further, the recombinant Vero cell line cultivation was scaled up to a 2000 L fully disposable bioreactor under optimized conditions, resulting in high viral titers, and ~1.0 × 10^7^ plaque forming units (PFU)/mL [154].

The FDA-approved process analytical technology (PAT) has now been rapidly employed in chimeric vaccine production to ensure real-time monitoring of production processes via cutting-edge tools and technologies. Recent innovations in configuration and robustness of sensor technology have allowed for the planning and analysis of, and control over the processes, such as bioreactor design, substrate concentration, the metabolic state and specific growth rate of cells, physiochemical changes, and the stability of recombinant products. Multivariate PAT tools in the application are on-line, in-line, at-line spectroscopy, including IR (infrared), NIR (near-infrared), Raman, fluorescence, and high-throughput MALDI (matrix-assisted laser desorption/ionization) mass spectrometer [155]. In recent work, using an insect-based baculovirus system, the PAT platform has been utilized to produce rabies virus-like particles. During the production process, the quantification of all stages of growth and other biochemical parameters were monitored via on-line and off-line Raman spectroscopy in association with a multivariate computational model for data analysis [156].

## 6. Downstream Process Development: Current Approaches and Future Trends

The downstream processing of chimeric vaccines is crucial because it directly contributes to product quality and its regulatory approvals for commercial use. The generalized downstream processing steps include cell harvesting, clarification, purification, and polishing. At the industrial scale, the cell harvesting or separation of culture media is facilitated via centrifugation techniques, while high-pressure homogenization and bead milling techniques mediate cell disruption. Furthermore, the purification and polishing steps combine conventional chromatography techniques and high-throughput technologies [157].

The steps of product recovery and downstream processes depend on protein localization. For instance, the intracellularly expressed inclusion bodies require cell disruption, followed by solubilization under denaturing conditions and refolding. The inclusion bodies are traditionally refolded using pulse dilution, column refolding, and diafiltration, followed by dialysis [158]. Recently, *E. coli*-produced anti-SARS-CoV-2 chimera was efficiently solubilized using 1% (*w*/*v*) anionic detergent N-lauroylsarcosine (sarkosyl). The removal of sarkosyl was facilitated by applying Amberlite XAD-4, a polymeric absorbent. The structural and functional assays further confirmed the stability and bioactivity of the chimeric protein [159]. Another study demonstrated that oxidized glutathione assisted in vitro refolding of tetravalent chimeric dengue viral antigen EDIIIT2, which requires proper disulfide bonds and native configuration for the immunogenic activity [160]. Amphipathic mono- and di-alcohols, such as 2-methyl-2,4-pentanediol (MPD), are also used to refold and assemble chimeric VLPs denatured by anionic detergent sodium dodecyl sulfate (SDS). However, the refolding efficiency depends on various factors, such as SDS/MPD ratio, protein concentration, temperature, pH, and ionic strength [161].

### 6.1. Chromatography Techniques: Conventional and Modern Approaches

Affinity, ion exchange, gel-filtration, and hydrophobic chromatography techniques are routinely applied to purify bioactive chimeric vaccines. Affinity chromatography selectively allows for the purification of recombinant antigens, vaccines, and viral vectors via frequently used affinity ligands, such as immunoaffinity ligands, immobilized metal ions, lectin, and heparin [162]. Recently, immobilized nickel ions have been utilized to purify the multi-epitope vaccine against *H. pylori* under denaturing conditions. The Western blot analysis further confirmed the successful purification of vaccine candidates constituting epitopes of urease (UreB), neutrophil-activating protein (Nap), and *H. Pylori* adhesin protein (HpaA) [163]. Single-step Ni-NTA (nickel-nitrilotriacetic acid) chromatography has also been used to purify C-terminally histidine-tagged chimeric vaccine construct constituting alpha toxin, necrotic enteritis toxin B-like (NetB) and TpeL toxin of *C. perfringens* [164]. A C-terminal affinity tag (C-tag) consisting of four amino acids, glutamic acid–proline–glutamic acid–alanine (E-P-E-A), was employed for the purification of *P. falciparum* reticulocyte-binding protein homolog 5 (PfRH5) expressed in ExpreS2 *Drosophila* S2 cell line. The potential antigen was purified using an affinity resin CaptureSelect^TM^ linked to a camelid single-chain antibody named NbSyn2. This novel affinity-based ligand allowed for the single-step purification of PfRH5 protein, yielding >85% recovery and >70% purity [165]. Small, stable, single-domain proteins known as scaffolds were also investigated to provide easy and robust affinity-based purification in compliance with the cGMP (current Good Manufacturing Practices) production process. Scaffold-based nanofitin protein coupled with commercially available Eshmuno^®^ beads resin was developed for the industrial-scale purification of a vaccine candidate against *Streptococcus pyogenes* [166]. Cryogel, based on the cryo-polymerization of 2-hydroxyethyl methacrylate has been designed for the affinity-based purification of DNA vaccine against influenza. Based on cryo-gelation, the stationary phase provides various advantages, such as a defined interconnected structure, higher binding capacity, and economical and easy purification [167].

Ion exchange chromatography (IEC) is extensively applied at the industrial scale to capture and purify virus-based chimeric vaccines. The use of ion exchangers helps in the successful removal of host cell proteins, contaminating proteins, DNA components, media components, toxins, and viral particles. In a recent study, a combination of IMAC (Immobilized Metal Affinity Chromatography) and anion-exchange-chromatography-purified chimeric vaccine candidate constituting the receptor binding domain (RBD) segment of SARS-CoV-2 delta and heat-labile enterotoxin B subunit of *E. coli* (LTB). The biologically active LTB-RBD chimeric antigen was recovered after a two-step purification strategy, indicating an easy and economical purification strategy for tagless chimeric antigens [168].

Hydrophobic interaction chromatography (HIC) purifies specific viral antigens under relative hydrophobicity, while size exclusion chromatography (SEC) or gel permeation separates proteins based on molecular sizes. In one study, the molecular surface hydrophobicity of chimeric VLPs was analyzed and mathematically calculated to design purification strategies based on the quality-by-design (QbD) approach. Several chimeric hepatitis B core antigen (HBcAg)-based VLPs of nuclear protein, matrix protein 2 of influenza A virus, and ovalbumin were successfully purified via HIC. The experimental analysis and mathematical calculations revealed that inserting these foreign epitopes in HBcAg-derived VLPs increased surface hydrophobicity. Thus, the preheat treatment, followed by HIC and SEC, were employed to purify chimeric VLPs with a higher recovery yield ranging from 33.2% to 43.7% [169]. The mammalian 293F cell-expressed chimeric bivalent HPV: HIV (L1:P18I10) vaccine was purified via a series of conventional purification techniques in cascade involving cation exchange chromatography, size exclusion chromatography, and heparin affinity chromatography. The novel purification technique improved the purity and recovery of the bivalent vaccine candidate by 38 fold and 6 fold, respectively, compared to conventional ultracentrifugation treatment [170]. A similar purification assembly set-up was also used to purify another chimeric VLP, consisting of T20 anti-fusion peptide of HIV1-gp41 fused into the L1 capsid protein of HPV [171]. In another study, a chimeric malaria-transmission-blocking vaccine expressed in *Lactobacillus lactis* was successfully purified via the combinatorial application of anion exchange and C- tag affinity-based chromatography, with a final product yield of ~12 mg/L corresponding to a 40% recovery rate [172]. In another approach, a combination of Q sepharose anion resin and CMM HyperCel mixed-mode cation exchange column was used to purify tagless fusion chimeric vaccine candidate PfMSPFu24, consisting of conserved regions of PfMSP-3 and PfMSP-1 merozoite surface antigens of *Plasmodium falciparum*. The fusion chimera was purified with >98% purity, as confirmed by the reversed-phase high-performance liquid chromatography [173]. Recently, chimeric hepatitis B core antigen (HBcAg)-based VLPs were purified to ~90% purity using a combination of Fractogel^®^EMD diethylaminoethyl (DEAE) ion exchanger and sepharose 4 fast flow (FF) gel exclusion column. The preS1 epitopes of hepatitis B surface antigen were displayed on the surface of HBcAg-based VLPs and aimed to elicit anti-preS1 immunogenicity and HBcAg-derived T-cell proliferation [174].

Gag-based VLPs of the HIV-1 virus form the basis for developing various chimeric vaccines and delivery agents. However, the Gag VLPs derived from animal cells are routinely subjected to conventional methods of purification, which are not scalable and reproducible. In a recent study, researchers demonstrated four-step purification schemes to yield Gag-based VLPs with higher purity and stability. Purification techniques involving dead-end filtration, QXT membrane adsorber IEC, sepharose 4 Fast Flow SEC, and lyophilization were used in cascade to remove host-cell impurities with recovery yields of around 23% [175]. In a similar study, HIV-1 Gag-based chimeric SARS-CoV-2 VLPs were purified via two clarification steps using depth filters, followed by anion exchange purification. The extended gel-permeation-mediated polishing step drastically removed the contaminating host cell proteins, yielding chimeric VLPs with improved purity of 31.3% [176]. Recently, weak anion diethylaminoethyl (DEAE) resin in combination with IMAC, has been utilized to obtain targeted purity (˃98%) of chimeric human papillomavirus-16 L1 capsid protein-based vaccine consisting of multi-epitopes from tetanus and diphtheria toxin [177].

Recently, multimodal or mixed-mode chromatography has been widely applied to purify chimeric vaccines. The mix-mode size-exclusion Capto^TM^ Core 700 resin of positively charged hydrophobic octylamine ligands has been successfully used to a purify chimeric VLP-based influenza vaccine. The porous bed of octylamine excluded impurities via the flow-through purification technique to give correctly assembled VLPs with a 2.4-fold increased purity and recovery yields of ~90%. Further, the removal of host cell proteins, including outer membrane protein F, was assisted by applying detergent extraction and a gel permeation polishing step to yield ~90% pure chimeric VLPs and ~20% recovery yield [178]. The Capto^TM^ Core 700 resin with combined size exclusion and ion exchange capabilities was also utilized for flow-through-based purification of large chimeric vesicular stomatitis virus with ~99% removal of host-cell proteins [179]. In another study, researchers used Fractogel^®^-trimethylammoniumethyl (TMAE) anion-based resin comprising a cross-linked polymethacrylate scaffold grafted with tentacles activated with the trimethyl aminoethyl anionic groups, to remove host cell proteins and achieve high-purity of the chimeric vaccine. The polymer grafted resin was used to purify insect cell-line-expressed chimeric VLPs constituting HIV-1 Gag capsid protein and the H1 hemagglutinin of influenza virus. The one-step purification via grafted anion exchanger removed around 94% of the total-process-related proteins, including contaminating baculovirus capsid proteins [180].

Various other emerging technologies of product recovery, like the aqueous-two phase extraction (ATPE) system, design of experiments (DoE), process analytical technology (PAT), and continuous downstream processing, are the future of cost-effective recovery of chimeric vaccines. The applications of these techniques in chimeric vaccine purification have been discussed in the following section.

### 6.2. Emergent Technologies in Product Recovery

The aqueous two-phase extraction (ATPE) system is an advanced purification platform based on liquid–liquid extraction. It includes multiple phase-forming systems, such as polymer/polymer, polymer/salt, ionic liquid, and alcohol-based ATPE [181]. Recently researchers have utilized various modified ATPE systems, including PEG/ammonium sulfate-based ATPE, continuous PEG12000/citrate ATPE, osmolyte-enhanced PEG12000/citrate ATPE, for the successful purification of recombinant HIV-based VLPs [182,183]. Various other modifications, including natural deep eutectic solvent (NADES)-based ATPE, are becoming robust, economical, and biocompatible alternatives for the recovery and purification of recombinant vaccines [184]. However, ATPE technology is currently in the developmental stage with promising recovery yields for chimeric vaccines.

Design of experiments (DoE) strategies have also been established to make the downstream process more convenient and efficient. The approach has been used to purify an *N. benthamiana*-expressed chimeric transmission-blocking malaria vaccine candidate constituting two antigens from the sexual stage (*Pfs*25, *Pfs*230) and an additional blood stage antigen (*Pf*Rh5) of *P. falciparum*. Blanching conditions were optimized to effectively remove ~90% of host cell proteins, while octyl sepharose was employed to purify stable proteins with >70% purity [185].

The real-time monitoring of downstream processes has become convenient via high-throughput process analytical technology, where current Good Manufacturing Practices (*cGMP*) are followed to obtain quick regulatory approvals. Tools such as third-generation spectrophotometers, high-performance liquid chromatography (HPLC), circular dichroism, etc., have been developed for on-line analysis of concentration, purity, and modifications present in purified protein. In addition, a novel, simple western technology that represents a fast and automated capillary-based western assay has been developed. The technique has been successfully utilized for at-line qualitative and quantitative assessment of SARS-CoV-2 spike (S) protein in the V590 vaccine candidate production against COVID-19 [186]. In another study, in-station real-time cell analysis assisted by dimensionless cell index has proven to be a robust alternative method to count real-time chimeric yellow fever dengue virus titers [187].

Continuous downstream processing at the industrial scale has various advantages, such as process intensification, automation, higher recovery yields, time-saving, and economical operations. Advanced disk stalk and tubular centrifuges are currently applied for continuous cell harvesting, whereas high-pressure homogenization and bead milling provide continuous cell lysis. The continuous filtration modes, including alternative tangential flow and single-pass tangential flow filtration, are evolving techniques that provide reduced aggregation and improved recovery yields of chimeric VLPs [188]. A recent report demonstrated the continuous and integrated high-throughput chromatographic purification of microbially expressed murine-polyomavirus-based chimeric VLPs. The high-quality chimeric VLPs, with a recovery yield of 88.6%, were purified via coupling flow-through Capto^TM^ Q chromatography with multimodal period counter-current chromatography technique. In the technology, varying inlet feed concentrations and applying salt-resistant mixed-mode resins allows for in-stream and economical purification with greater recovery yields [189]. Hillebrandt et al. [190] reported a 2.6-fold enhancement in productivity yields of chimeric hepatitis B core antigen (HBcAg) VLPs via integrating capture and purification steps. A cross-flow filtration-based set-up was assembled in-stream with multimodal Capto^TM^ Core 400 size exclusion chromatography to yield higher product yields of around 86%.

## 7. Conclusions

The review provides insight into the various limitations associated with conventional vaccines and an urgent need to develop prophylactic chimeric vaccines. These vaccines can provide comprehensive immunization against multiple pathogens in a single formulation. Omics-based research is widely applicable in designing chimeric vaccines. Recent advancements in bioinformatics tools have allowed researchers to develop highly immunogenic and stable chimeric antigens. Furthermore, the industrial-scale production of chimeric vaccines requires a combination of traditional and modern technologies. Several successful vaccines have been produced using conventional upstream and downstream processes. However, bioengineering and high-throughput technologies in process development have created new opportunities to develop difficult-to-express chimeric proteins. The review summarized numerous challenges associated with conventional process development while outlining new strategies in upstream and downstream processes.

## Figures and Tables

**Figure 1 vaccines-11-01828-f001:**
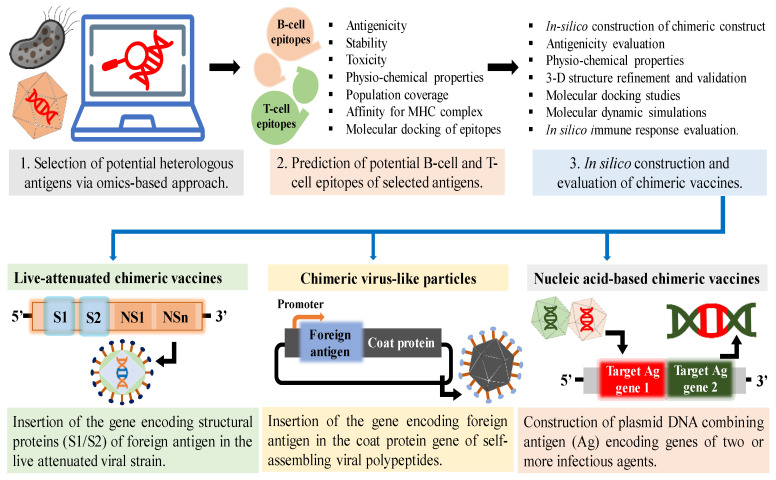
Development of chimeric vaccines using in silico approach.

**Figure 2 vaccines-11-01828-f002:**
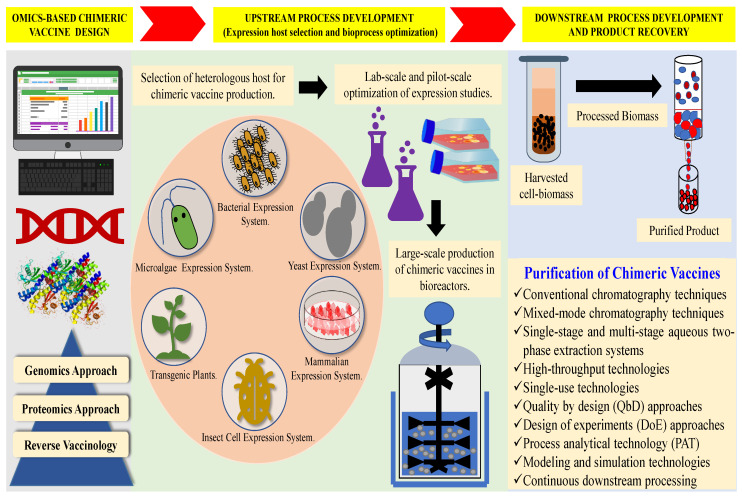
Schematic representation of the different stages of chimeric vaccine development.

**Table 1 vaccines-11-01828-t001:** List of commercially approved recombinant vaccines developed in different expression hosts targeted against human diseases.

Commercial Vaccines	Target	Protective Antigen/Formulation	Expression Host	Manufacturer	Reference
Recombivax HB^®^	Hepatitis B virus	Hepatitis B surface antigen (HBsAg)/Subunit	*S. cerevisiae*	Merck & Co., Inc.,NJ, USA	[13]
Engerix^®^-B	Hepatitis B virus	Hepatitis B surface antigen (HBsAg)/Subunit	*S. cerevisiae*	GlaxoSmithKline, Brentford, UK	[14]
GeneVac-B	Hepatitis B virus	Hepatitis B surface antigen (HBsAg)/Subunit	*H. polymorpha*	Serum Institute of India Pvt. Ltd., Pune, India	[15]
Hepacare^®^	Hepatitis B virus	Triple antigen (S, pre-S1, and pre-S2)/Subunit	Mammalian cell line	Medeva Pharma Plc, Speke, UK.	[16]
Twinrix	Hepatitis A and Hepatitis B virus	Inactivated hepatitis A virus (strain HM175) and non-infectious HBsAg/Bivalent subunit	*S. cerevisiae*	GlaxoSmithKline, Brentford, UK	[17]
HBVaxPRO^®^	Hepatitis B virus	Hepatitis B surface antigen (HBsAg)/Subunit	*S. cerevisiae*	Sanofi Pasteur MSD, Lyon, France	[18]
LYMErix^®^	*Borrelia burgdorferi*	Lipoprotein OspA/Subunit	*E. coli*	GlaxoSmithKline, Brentford, UK	[19]
Trumenba^®^	*Neisseria meningitidis* serogroup B	Recombinant factor H binding protein (fHbp), from subfamily A (A05) and subfamily B (B01)/Subunit	*E. coli*	Pfizer Inc., New York, USA	[20]
Bexsero^®^	*Neisseria meningitidis* serogroup B	Outer membrane vesicle protein, fHbp fusion protein, NadA protein, and NHBA fusion protein/Multicomponent recombinant	*E. coli*	GlaxoSmithKline, Brentford, UK	[21]
Gardasil^®^	Human papillomavirus (HPV)	HPV (Types 6, 11, 16, and 18) L1 protein/VLPs	*S. cerevisiae*	Merck & Co., Inc.,NJ, USA	[22]
Gardasil^®^ 9	Human papillomavirus (HPV)	Nine-valent HPV (Types 6,11,16,18,31,33,45,52,58) L1 protein/VLPs	*S. cerevisiae*	Merck & Co., Inc.,NJ, USA	[23]
Cecolin^®^	Human papillomavirus (HPV)	HPV (Types 16,18) L1 protein/VLPs	*E. coli*	Xiamen Innovax Biotech Co., Ltd., Xiamen, China	[24]
Mosquirix^TM^ (RTS,S/AS01)	*Plasmodium falciparum* (Malaria)	Chimeric protein: RTS in fusion with HBsAg/VLPs	*S. cerevisiae*	GlaxoSmithKline, Brentford, UK	[25]
Shingrix	Varicella zoster virus (Shingles)	Glycoprotein E antigen (gE)/Subunit	CHO cell line	GlaxoSmithKline, Brentford, UK	[26]

**Table 2 vaccines-11-01828-t002:** List of chimeric and recombinant vaccine candidates expressed in *Pichia pastoris*.

Pathogens	Target Antigen/Vector/Host Strain	Strategy and Outcome	References
**Chimeric vaccines:**			
Dengue virus (DENV)	HBcAg-EDIII-2/pPICZ A/*P. pastoris* KM71H strain.	Chimeric vaccine developed displaying the envelope domain III of dengue virus type-2 (DENV-2) on hepatitis B core antigen (HBcAg)-based VLPs. Insertion of EDIII-2 in the c/e1 loop of HBcAg yielded structurally and functionally stable chimeric VLPs. A modest titer of DENV-2-specific virus-neutralizing antibodies was found in the mice immunized with chimeric VLPs.	[76]
Dengue virus (DENV)	rEDIII-T/pPIC9K vector/*P. pastoris* GS115 (*his4*) strain.	Tetravalent chimeric vaccine developed via fusion of receptor-binding envelope domain III of four serotypes DENV-1, DENV-2, DENV-3, and DENV-4 via pentaglycyl peptide linkers. Neutralizing activity against infectivity of all four serotypes of the dengue virus was obtained in the BALB/c mice model. The in vitro studies revealed Th2 (T helper 2) immune response activation.	[77]
Dengue virus (DENV)	Bivalent antigen/pAO815 vector/*P. pastoris* GS115 (*his4*) strain.	Co-expression and co-assembly of envelope (E) proteins from DENV-1 and DENV-2 into bivalent mosaic chimeric VLPs known as mE1E2_bv_ VLPs. Serotype-specific virus-neutralizing antibodies obtained in BALB/c mice model and lack of antibody-mediated antibody-dependent enhancement (ADE) activity in vivo model.	[78]
Zika virus (ZIKV)	ZIKV EDIII/pAO815 vector/*P. pastoris* GS115 *(his4)* strain.	Chimeric VLPs developed by co-expression and co-purification of ZS (envelope domain III of Zika virus fused with hepatitis B surface antigen) and S (hepatitis B surface antigen) in a 1:4 ratio. Chimeric VLPs were found to be immunogenic in BALB/c mice model, capable of neutralizing ZIKV particles.	[79]
*Toxoplasma gondii*	SAG1/2/pPICZα A vector/*P. pichia* X-33 strain.	Recombinant chimeric surface antigens 1 and 2 (SAG1/2) of *T. gondii* in fusion with C-terminal-his tag was expressed and purified via Ni-NTA affinity chromatography. Vaccination via immunogenic chimeric protein exhibited SAG1- and SAG2-specific immune response in BALB/c mice model and the secretion of IFN-γ, the Th1 (T helper 1)-specific cytokines.	[80]
*Mycobacterium avium* subsp. *paratuberculosis* (MAP)	HBHA-FAP-P/pPICK9K vector/*P. pastoris* strain GS115 strain.	Expression of the chimeric fusion protein consisting of heparin-binding hemagglutinin (HBHA) adhesin protein and the antigenic region of the fibronectin attachment protein-P (FAP-P) of *Mycobacterium avium* subsp. *paratuberculosis*. Post-translationally methylated HBHA protein in fusion with FAP-P developed similarly to the native HBHA in 3D structure.	[81]
Human papillomavirus (HPV)	Chimeric HPV-16 LI-L2 protein/pBLHIS–IX/*P. pastoris* KM71 strain.	Chimeric protein known as ChiΔF-L2, developed via replacing the h4 helix of HPV-16 L1 major capsid protein by the 108–120 amino acids corresponding to L2 minor capsid protein. Successfully produced ~23.61 mg/L chimeric HPV L1/L2 protein during fed-batch fermentation.	[82]
*Plasmodium vivax*	PvAMA1_66_-MSP1_19_/pPIC9K vector/*Pichia pastoris* GS115 strain.	Chimeric antigen developed, consisting of the immunogenic apical membrane antigen 1 (AMA-1) ectodomain and the C-terminal region of the merozoite surface protein 1 (MSP-1) of *P. vivax*. High anti-AMA-1, anti-MSP-1 antibody titers and the induction of IFN-gamma secreting immune cells in immunized mice model.	[83]
*Leishmania donovani* (Visceral leishmaniasis)	Synthetic gene (LeiSp)/pPICZA vector/*P. pastoris* X-33 strain.	Reverse vaccinology was implemented to develop a chimeric LeiSp multi-epitope vaccine candidate consisting immunogenic membrane and secretory proteins of *L. donovali infantum* and salivary proteins of *P. argentipes* sand fly vector. Significant upregulation of proinflammatory cytokines and decreased anti-inflammatory cytokines in THP1 cell line.	[84]
**Recombinant vaccines:**			
Chikungunya virus (CHIKV)	Structural capsid and envelope (E3, E2, 6K, E1) protein/pPIC9K vector/*Pichia pastoris* GS115 strain.	Structural and capsid protein of CHIKV expressed and purified via ultrafiltration using discontinuous sucrose gradient. CHIKV-VLPs induced both cellular and humoral immune responses in mice models. Positive in vitro and in vivo neutralization activity of anti-CHIKV-VLPs antibodies.	[85]
Japanese encephalitis virus (JEV)	Modified JEV prM/Env gene/pPICZA vector/*Pichia pastoris* X-33 strain.	Expression and purification of JEV prM/ENV proteins via Ni-NTA chromatography followed by an in vitro assembly of JEV-VLPs. Robust cellular and humoral immune response in mice models. After immunization, a high titer of neutralizing antibodies against JEV was obtained in the pig model.	[86]
Zika virus (ZIKV)	ZIKV Env and NS1 proteins/pPGKΔ3α vector /*P. pastoris* GS115 strain.	Immunogenic epitopes ZIKV envelope (E) and NS1 proteins displayed on the surface of *P. pastoris* via GPI anchored AG-α1 protein. The in vitro immunogenic activity of recombinant yeast indicated by the activation of the lymphocytes and monocytes of mouse spleen.	[87]
Dengue virus (DENV)	ENVIII-2 gene/pPICZA vector/*P. pastoris* KM71H strain.	Intracellular expression of dengue Virus type-2 envelope domain III. Elevation of virus-neutralizing antibodies in BALB/c mice model estimated by plaque-reduction neutralization test.	[88]
Hepatitis E virus	ORF-2/pPICZα vector/*P. pastoris* KM71H strain.	Gene-encoding 112–608 aa of capsid forming ORF-2 of the hepatitis E virus expressed and purified as potential virus-like particles. Generation of ORF-2-specific IgG antibody and splenocyte proliferation in BALB/c mice model.	[89]
Polio virus	PV-3 SC8/pPink HC vector/PichiaPink^TM^ strain.	Thermally stable PV-3 SC8 VPLs developed under dual AOX1 regulation, consisting structural precursor protein (P1) and viral protease (3CD). Wild-type PV-3 VLPs elicited high C and D antigenicity whereas, thermally stable PV-3 SC8 VLPs exhibited more D antigens.	[90]
Dengue virus (DENV)	DENV-1 EDIII/pPICZA vector/*P. pastoris* KM71H strain.	Codon-optimized gene E glycoprotein (ectodomain III) of DENV-1 expressed and purified as self-assembled VLPs. DENV-1 E VLPs elicited virus-serotype-specific neutralizing antibodies in mice model.	[91]
Dengue virus (DENV)	DENV-4 EDIII/pPICZ-A vector/*P. pastoris* KM71H strain.	Codon optimized gene DENV-4 EDIII expressed and purified as VLPs via Ni-NTA affinity chromatography under denaturing conditions. DENV-4 E VLPs generated immunogenic response and EDIII-derived virus serotype-specific neutralizing antibodies in mice models.	[92]

**Table 3 vaccines-11-01828-t003:** List of commercially approved human and veterinary vaccines developed using an insect cell expression system.

Disease/Pathogen	Commercial Vaccine	Protective Agent/Vaccine Type	Insect Cells Lines	Manufacturer	References
**Vaccines for human-use**					
Cervical Cancer (HPV)	Cervarix^®^	L1 protein/VLP	High Five Cells	GlaxoSmithKline, Brentford, UK	[110]
Prostate Cancer	Provenge^®^	PAP-GM-CSF Fusion protein/Cellular vaccine	Sf-21 cells	Dendreon Corporation, California, USA	[111]
Influenza Virus	Flublok^®^	HA glycoprotein/Subunit	expresSF+ cells	Sanofi Pasteur, Paris, France	[112]
Influenza Virus	FlublokQuadrivalent^®^	HA quadrivalent/Subunit	expresSF+ cells	Sanofi Pasteur, Paris, France	[113]
COVID-19	Nuvaxovid/ Covovax^TM^	Spike S protein/Spike protein nanoparticle	Sf9 cells	Novavax, Malvern, PA, USA/ Serum Institute of India, Pune, India	[114,115]
**Vaccines for veterinary use**					
Classical Swine Flu Fever Virus	Porcilis Pesti^®^	E2 protein/Subunit	Sf-21 cells	Merck & Co., Inc., NJ, USA	[116]
Classical Swine Flu Fever Virus	Bayovac^®^ CSF E2	E2 protein/Subunit	Sf-21 cells	BAYER AG/Pfizer Animal Health (Nordrhein-Westfalen, Germany/Groton, CT, USA (Discontinued)	[109]
Porcine Circovirus Type 2 (PCV-2)	Porcilis^®^ PCV	PCV2a ORF2 protein/VLP	Tn5 cells	Merck & Co., Inc., NJ, USA	[117]
Porcine Circovirus Type 2 (PCV-2)	IngelVac CircoFLEX^®^	PCV2a ORF2 protein/VLP	Tn5 cells	B. Ingelheim, Berlin, Germany	[118]
Porcine Circovirus Type 2 (PCV-2)	Circumvent^®^ PCV2	PCV2a ORF2 protein/VLP	Tn5 cells	Merck & Co., Inc., NJ, USA	[119]
Porcine Circovirus Type 2 (PCV-2)	CircoGard^®^	PCV2b ORF2 protein/VLP	(Not specified)	Pharmgate Animal Health, Wilmington, North Carolina, USA	[120]
Porcine Parvovirus (PPV)	ReproCyc^®^ ParvoFLEX	PPV 27a VP2/VLP	(Not specified)	B. Ingelheim, Berlin, Germany	[121]

## Data Availability

The data presented in this study are available in referenced articles.

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
