# Peer review of "Challenges and Opportunities in the Process Development of Chimeric Vaccines"

_vaccines, 2023, doi:10.3390/vaccines11121828_

Round 1

Reviewer 1 Report

This review provides a comprehensive analysis of the obstacles and promising new directions in the bioprocess development of chimeric vaccines. The author has done the difficult task of merging the mass-scale recombinant protein and peptide expression with chimeric vaccine design and their production.  The review discusses the latest updates on chimeric vaccine design,  different expression systems for mass-scale production of the chimeric vaccine, and their recovery process. The data presented in the study is beneficial to academicians and industry persons. 

Few very minor  changes which may be considered.

Line no. 42 "The killed or inactivated ...... radiations," may be reframed to bring clarity.

Page 4, Line 27: The font size of "P. falciparum NF54" should be reduced to match the surrounding text.

Page 4, Line 132: The font size of "Neisseria meningitides" should be reduced to match the surrounding text.

Page 36, Lines 313-317: The text is shaded with different backgrounds unnecessarily that may be removed.

Table 3: Few pieces of text in the table are shaded with different backgrounds for no apparent reason that may be removed

Author Response

Reviewer-1

First of all, I must endorse the reviewers' and Editor's sincere efforts in reviewing our manuscript. We appreciate the constructive criticism, suggestions, and comments. Keeping in view of the reviewer's suggestion, the manuscript has been comprehensively revised, and all suggestions have been incorporated. Major changes have been highlighted in yellow. The English language writing and grammar are thoroughly checked and improved.

Question: Line no. 42 "The killed or inactivated ...... radiations," may be reframed to bring clarity.

Answer: As suggested, the sentence has been reframed to bring clarity.

Page 4, Line 27: The font size of "P. falciparum NF54" should be reduced to match the surrounding text.

Answer:. As suggested, a required change in the font size has been made.

Page 4, Line 132: The font size of "Neisseria meningitides" should be reduced to match the surrounding text.

Answer: As suggested, a required change in the font size has been made in the revised manuscript.

Page 36, Lines 313-317: The text is shaded with different backgrounds unnecessarily that may be removed.

Answer: As suggested, editing has been done to address the issue.

Table 3: Few pieces of text in the table are shaded with different backgrounds for no apparent reason that may be removed

Answer: As suggested, the editing has been done to remove shading.

Reviewer 2 Report

Comments for the authors

In view of the outbreaks of dreadful diseases have led researchers to develop economical chimeric vaccines that can cater to a large population in a shorter time, the authors introduced omics-based vaccine development approaches, upstream process development for chimeric vaccines and downstream process development (current approaches and future trends) . and the authors emphasized that owing to the complex structures and complicated bio-processing of evolving pathogens, various high-throughput process technologies have come up with added advantages. Recent advancements in high-throughput tools, process analytical technology (PAT), quality-by-design (QbD), design of experiments (DoE), modeling and simulations, single-use technology, and integrated continuous bioprocessing have made scalable production more convenient and economical.

Major concerns

1.        The authors should define the "Chimeric vaccines" and compare to non-chimeric vaccines.

2.        what is the general advantages or the strategies(design) of chimetic vaccines.

3.        What is difference of chanllenges and opportunities in the process develp[ment between the chimeric vaccines and non-chemeric vaccine.

Minor concerns

1.        Line 149, the title “2. Omics-Based Vaccine Development Approaches” modified as “2. Omics-Based Chimeric Vaccine Development Approaches” may be better.

2.        line 822, “next-gen” should be “next-generation”.

Author Response

Reviewer-2 

First of all, I must endorse the reviewers' and Editor's sincere efforts in reviewing our manuscript. We appreciate the constructive criticism, suggestions, and comments. Keeping in view of the reviewer's suggestion, the manuscript has been comprehensively revised, and all suggestions have been incorporated. Major changes have been highlighted in yellow. The English language writing and grammar are thoroughly checked and improved.

Major concerns

Question 1: The authors should define the "Chimeric vaccines" and compare to non-chimeric vaccines.

Answer: The manuscript has been comprehensively revised, and the following new sections have been included in the revised manuscript to compare chimeric and non-Chimeric vaccines.

  • Section No. 2: Chimeric Vaccines: Future of Vaccinology with State-of-the-Art Technologies
  • Subsection 2.1: Strategies to design Chimeric vaccines
  • Subsection 2.2: Types of chimeric vaccines

Several examples have also been discussed, citing the added advantages of chimeric vaccines.

Question 2: What are the general advantages or the strategies(design) of chimeric vaccines?

Answer: The general advantages and strategies have also been included in:

Subsections: 2.1: Strategies to design chimeric vaccines”

Subsections: 2.2: Types of chimeric vaccines

Question 3: What is difference of challenges and opportunities in the process development between the chimeric vaccines and non-chimeric vaccine.

Answer: The manuscript has been revised by including one new section, i.e.section 4, describing new challenges and opportunities in the bioprocess development of chimeric vaccines.

 Minor concerns

Question 1:  Line 149, the title “2. Omics-Based Vaccine Development Approaches” modified as “2. Omics-Based Chimeric Vaccine Development Approaches” may be better.

Answer: As suggested, the changes have been incorporated into the revised manuscript

Question2 : line 822, “next-gen” should be “next-generation”.

Answer: As advised, the suggestion has been incorporated into the revised manuscript.

Reviewer 3 Report

The topic is interesting and the review is a laborious work. But the review is somehow hard to read through and seems to have many issues to be published in vaccines.

1. What authors mean “chimeric vaccines” is not clear. The manuscript seems to review protein vaccines in general, mainly focused on VLPs, instead of chimeric vaccines.

2. The topic is too broad. It is better to focus on particular aspects on the topic.

For example, authors reviewed “upstream process development”. But the topic is too broad. Selection of media components, cultivation mode, or bioreactor type, etc. depends on the vaccine platform.

4. In this manuscript many abbreviations are used without explanation, especially in “downstream process development” section. The explanation of each reference is too compact to figure out.

5. There are many careless mistakes. For example, usage of capital and small letters is not unified.

Minor points.

Line 370. RBD is not a protein name. RBD is a domain in spike protein.

Table 1. “pappiloma” reads “papilloma”. 

The topic is interesting and the review is a laborious work. But the review is verbose and hard to read through and has many issues to be published in vaccines.

Author Response

Reviewer-3:

First, we must sincerely endorse the reviewers' and Editor's efforts in reviewing our manuscript. We appreciate the constructive criticism, suggestions, and comments. Keeping in view of the reviewer's suggestion, the manuscript has been comprehensively revised, and all suggestions have been incorporated. Major changes have been highlighted in yellow. The English language writing and grammar are thoroughly checked and improved.

Major concerns

Question 1: What authors mean “chimeric vaccines” is not clear. The manuscript seems to review protein vaccines in general, mainly focused on VLPs, instead of chimeric vaccines.

Answer: The manuscript has been comprehensively revised, and the following new sections have been included in the revised manuscript to compare chimeric and non-Chimeric vaccines.

  • Section No. 2: Chimeric Vaccines: Future of Vaccinology with State-of-the-Art Technologies
  • Subsection 2.1: Strategies to design Chimeric vaccines
  • Subsection 2.2: Types of chimeric vaccines

Several examples have also been discussed, citing the added advantages of chimeric vaccines.

We tried to review the available literature on proteins, domains, potential antigens and VLPs-based chimeric vaccines. The use of VLPs-based chimeric vaccine has been recently tested and marketed. However, protein-based chimeric vaccines against multidrug resistance and highly infectious pathogens agents are also in the developmental stage with encouraging outcomes from clinical trials.

    Question 2: The topic is too broad. It is better to focus on particular aspects on the topic.

For example, authors reviewed “upstream process development”. But the topic is too broad. Selection of media components, cultivation mode, or bioreactor type, etc. depends on the vaccine platform.

Answer: Authors are thankful to the reviewer for raising this point. The production of chimeric vaccines has opened different challenges, like upstream antigen designing, bioprocess development, and product recovery. All these strategies have different bottlenecks and must be addressed to reduce production costs. Therefore, we have included different challenges, opportunities, and possible solutions to provide a wholesome understanding of the bioprocessing of chimeric vaccines available in literature. Recent advancements in high-throughput tools, process analytical technology (PAT), quality-by-design (QbD), design of experiments (DoE), modeling and simulations, single-use technology, and integrated continuous bioprocessing have been extensively reviewed to give an insight into various challenges associated with biomass cultivation, process optimization, and product recovery.

Question 3: In this manuscript, many abbreviations are used without explanation, especially in the “downstream process development” section. The explanation of each reference is too compact to figure out.

Answer: As pointed out, the concern has been addressed by incorporating all the suggestions in the revised manuscript.

Question 4: There are many careless mistakes. For example, the usage of capital and small letters is not unified.

Answer: As advised, the non-uniformity of a few words has been addressed.

Minor points.

Line 370. RBD is not a protein name. RBD is a domain in spike protein.

Answer: As advised, it has been corrected.

Table 1. “pappiloma” reads “papilloma”. 

Answer: Yes, it is a typographic mistake, and it has been addressed.

The English language writing and grammar are thoroughly checked and improved. New sections have been added and highlighted. However, general english and grammar editing are not highlighted in the revised manuscript.

Reviewer 4 Report

The review is not really focused on chimeric vaccines as the tittle would suggest, and is very superficial on the various topics it addresses.

English usage would need an extensive revision, including the tense used in some verbs and the usage of some words such as efficacy versus effectiveness or “cultivable” vs “culturable”.

Author Response

Reviewer-4:

First, we must sincerely endorse the reviewers' and Editor's efforts in reviewing our manuscript. We appreciate the constructive criticism, suggestions, and comments. Keeping in view of the reviewer's suggestion, the manuscript has been comprehensively revised, and all suggestions have been incorporated. Major changes have been highlighted in yellow. The English language writing and grammar are thoroughly checked and improved.

Question: The review is not really focused on chimeric vaccines as the tittle would suggest, and is very superficial on the various topics it addresses.

Answer:  We did not get any specific comments; however, general comments have been addressed in the revised manuscript. The industrial-scale production of chimeric vaccines requires a combination of traditional and modern technologies. Several successful vaccines have been produced using conventional upstream and downstream processes. However, bioengineering and high-throughput technologies in process development have created new opportunities to develop difficult-to-express chimeric proteins. Furthermore, we humbly submit that the central theme of the review is “Process Development of Chimeric Vaccines”. Therefore we have extensively discussed various opportunities and challenges in Process development (Section 5.2.1 to 5.2.6) and downstream processing (section 6.1 to 6.7).

In light of reviewer comments, the manuscript has been comprehensively revised, and the following new section has been included or extensively revised in the revised manuscript.

  • Section No. 2: Chimeric Vaccines: Future of Vaccinology with State-of-the-Art Technologies
  • Subsection 2.1: Strategies to design Chimeric vaccines
  • Subsection 2.2: Types of chimeric vaccines have been added to define chimeric vaccines and their added advantages.
  • Section 4: Describing new challenges and opportunities in chimeric vaccine development (comprehensively revised in the revised manuscript)
  • The English language writing and grammar are thoroughly checked and improved. New sections have been added and highlighted. However, general english and grammar editing are not highlighted in the revised manuscript.

Round 2

Reviewer 3 Report

  The revised manuscript is still hard to read through. Major concerns are not resolved. The topic is too broad and too long and not well organized.

New section No. 2 was added, but definition of “chimeric vaccines” is not clear. It seems too broad concept, including multi-epitope vaccine and hybrid vaccine, etc. Authors stated that they compare chimeric and non-chimeric vaccines in the new section, I only found the conceptual comparison, not the detailed one.

The flow of text is not clear. For example, The meaning to show Table 1 is not clear. In the text, authors described nucleic acid-based recombinant vaccines (lines 93-104). Then, authors abruptly discuss Table 1 (approved recombinant vaccines, not nucleic acid-based vaccines). 

There are still many careless mistakes. The examples are shown below.

Line 92. “human immunomodulatory virus” reads “human immunodeficiency virus”.

Lines 140 and 145. “cytolytic T lymphocytes (CTLs)” (l. 140) and “cytotoxic T lymphocyte (CTL)” (l. 145). The wording should be unified.

Line 471 and 476. “HBsAg (hepatitis B surface antigen)” (l. 471) and “Hepatitis B surface antigen (HBsAg)” (l. 476). The wording should be unified.

Author Response

Response to Reviewer’s comments

Manuscript Number: Vaccines-454184

Titled Challenges and Opportunities in the Process Development of Chimeric Vaccines

Question 1: The revised manuscript is still hard to read through. Major concerns are not resolved. The topic is too broad and too long and not well organized.

Answer1 :

  • First, I must sincerely endorse the reviewers' and Editor's sincere efforts in reviewing our manuscript. We appreciate the constructive criticism, suggestions, and comments. Keeping in view of the reviewer's suggestion, the manuscript has been comprehensively revised, and all suggestions have been incorporated. The English language writing and grammar are thoroughly checked and improved.
  • As per the suggestion, in the revised and restructured manuscript, we have significantly abridged the manuscript and reduced the number of references from 274 to 192.
  • We have removed a few sections and extensively revised the manuscript to specifically focus on the development of chimeric vaccines and their process optimization.
  • As per reviewer suggestions, we have majorly restructured the upstream and downstream process development sections as follows:

Section 5: Emergent Technologies in Upstream Process Development

Section 6.2: Emergent Technologies in Product Recovery

Question 2: New section No. 2 was added, but definition of “chimeric vaccines” is not clear. It seems too broad concept, including multi-epitope vaccine and hybrid vaccine, etc. Authors stated that they compare chimeric and non-chimeric vaccines in the new section, I only found the conceptual comparison, not the detailed one.

Answer 2: We have extensively revised Section 2 to define the chimeric vaccine with new examples. To better understand the concept, we also incorporated one new figure (Figure 1) illustrating strategies to develop chimeric vaccines using in silico approaches. In the later part of the review, the process development and product recovery strategies emphasizing different challenges and the accessibility of new technologies for chimeric vaccines in research are comprehensively reviewed.

Question 3: The flow of text is not clear. For example, The meaning to show Table 1 is not clear. In the text, authors described nucleic acid-based recombinant vaccines (lines 93-104). Then, the authors abruptly discuss Table 1 (approved recombinant vaccines, not nucleic acid-based vaccines). 

Answer 3: The authors agree with the reviewer's opinion; therefore, sections 1 and 2 are reorganized to maintain the flow for a better understanding of the manuscript.

As suggested, the incorporation of tables and figures in the revised manuscript is done for better clarity and understanding.

Question 4:

Line 92. “human immunomodulatory virus” reads “human immunodeficiency virus”.

Lines 140 and 145. “cytolytic T lymphocytes (CTLs)” (l. 140) and “cytotoxic T lymphocyte (CTL)” (l. 145). The wording should be unified.

Line 471 and 476. “HBsAg (hepatitis B surface antigen)” (l. 471) and “Hepatitis B surface antigen (HBsAg)” (l. 476). The wording should be unified.

Answer 4: we tender our apology for these avoidable mistakes. In the revised manuscript, we put our best efforts not to repeat the same.